# Genome-wide profiling of alternative splicing genes in hybrid poplar (*P.alba×P.glandulosa* cv.84K) leaves

**Ruixue Wang**[1,2☯], **Peng Yin**[1,2☯], **Yang Ruixia**[2], **Xiao Liu**[1,2], **Lie Luo**[2], **Jichen Xu**[1,2]*

**1** Beijing Advanced Innovation Center for Tree Breeding by Molecular Design, Beijing Forestry University, Beijing, China, **2** National Engineering Laboratory for Tree Breeding, Beijing Forestry University, Beijing, China

☯ These authors contributed equally to this work.
* jcxu282@sina.com

**Data Availability Statement:** All files are available from the NCBI database (accession numbers SRR12280784-SRR12280783).

## Abstract

Alternative splicing (AS) is a post-transcriptional process common in plants and essential for regulation of environmental fitness of plants. In the present study, we focus on the AS events in poplar leaves to understand their effects on plant growth and development. The hybrid poplar (*P.alba×P.glandulosa* cv.84K) leaves were collected for RNA extraction. The extracted RNA was sequenced using on an Illumina HiSeq™ 2000 platform. Using the *Populus trichocarpa* genome as the reference, a total of 3810 AS genes were identified (9225 AS events), which accounted for 13.51% of all the expressed genes. Intron retention was the most common AS event, accounting for 43.86% of all the AS events, followed by alternative 3′ splice sites (23.75%), alternative 5′ splice sites (23.71%), and exon skipping (8.68%). Chromosomes 10 had the most condensed AS events (33.67 events/Mb) and chromosome 19 had the least (12.42 events/Mb). Association analysis showed that AS in the poplar leaves was positively correlated with intron length, exon number, exon length, and gene expression level, and was negatively correlated with GC content. AS genes in the poplar leaves were associated mainly with inositol phosphate metabolism and phosphatidy-linositol signaling system pathways that would be significant on wooden plant production.

## Introduction

Most plant genes contain introns with conserved nucleotides (GT-AG) at both ends. The initial transcription product, pre-RNA, is spliced to remove introns and the remaining exons are ligated to form mature coding mRNA sequences. Therefore, accurate splicing of gene transcripts is essential for the growth and development of plants and is helpful for their adaptation to variable environment conditions [1].

Also, the pre-mRNA sequences can undergo alternative splicing (AS) to produce different splicing isoforms. For example, the circadian clock associated gene *CCA1* in *Arabidopsis* had two AS products, a full-size *CCA1α* isoform and an incomplete *CCA1β* isoform in which the MYB DNA-binding motif is missing [2]. Over-expression of *CCA1α* increased the plant's

**Funding:** The National Natural Science Foundation of China (#31672189) and Beijing Forestry University Undergraduate Training Program for Inovation and Entrepreneurship (#202010022062). The funders had no role in study design, data collection and analysis, decision to publish, or preparation of the manuscript.

**Competing interests:** The authors have declared that no competing interests exist.

freezing tolerance, whereas over-expression of *CCA1β* caused plants sensitive to freezing. β-hydroxyacyl ACP dehydratase is the key enzyme in the fatty acid synthesis pathway. In *Picea mariana*, the β-hydroxyacyl ACP dehydratase gene transcript retained intron 1 under normal temperature conditions and was not translated, whereas at temperatures below freezing, the normal transcript was present and was translated into the active enzyme [3]. In *Arabidopsis*, introns 2 and 3 of the disease resistance protein gene *RPS4* were shown to be crucial for the pathogen defense response. When these two introns were deleted the gene encoded a resistance protein with the complete TIR-NBS-LRR structure, but it had no disease resistance activity [4]. In tomato, AS of a *MLO* gene can lead to aberrant mRNA isoforms that cause the loss-of-function [5]. Transcriptome analysis in the plant–fungus interaction showed that 31% of the reads are indicative of putative alternative transcriptional events. GO-enrichment analysis revealed them significant involving in response to stimulus, regulation of metabolic process, carboxylesterase activity, and oxigen binding [6]. In poplar, the transcripts from leaf, root, xylem, and phloem of *Populus alba* var. *pyramidalis* were analyzed. The AS events occurred in 7,536 genes, of which 4,652 genes had multiple AS events [7]. *Comparably*, the isochorismate synthase (ICS) *gene* underwent extensive AS events in poplar that was rare in *Arabidopsis*, suggesting alternative splicing of *ICS* evolved independently in *Arabidopsis* and *Populus* in accordance with their distinct defense strategies. Besides, AS was also studied in *Arabidopsis* [8], *Eucalyptus* [9], *Picea abies* and *Pinus taeda* [10]. Therefore, inaccurate AS can produce abnormal transcripts that encode varied proteins with functional diversity, which could result in an effective post-transcriptional regulatory pattern [11]. Further, AS has the evolutionary characters of economy, rapidity, and low risk.

In this study, we aimed to dissect the AS events in leaves of the hybrid 84K poplar (*Populus alba* × *Populus glandulosa*), which has the advantages of easy rooting, short seedling period, and strong resistance to abiotic stress. The results will exhibit variations of AS events among poplar varieties and further contribute to a better understanding of the roles of AS in leaves on hybrid growth and development.

## Materials and methods

### Plant materials and growth

The seedlings of 84K poplar were grown in vermiculite mixed with nutrient soil at 1:1. The growth chamber conditions are at 25°C and in an 8 h dark/16 h light cycle. The mature leaves of well-growing plants in two months were collected and used for total RNA extraction.

### Isolation of total RNA

The total RNA was isolated from the poplar leaves using Trizol (Aidlab Bio Co Ltd, China) in accordance with the manufacturer's protocol. The residual DNA was removed with RNase-free DNase I (Takara Bio Inc, Japan) treatment for 30 min at 37°C. RNA quality was initially evaluated using an agarose gel and NanoDrop2000 spectrophotometer (Thermo Scientific, Waltham, USA). RNA integrity was assessed using the RNA Nano 6000 Assay Kit of the Agilent Bioanalyzer 2100 system (Agilent Tech, USA). Three replicates were conducted.

### Construction of cDNA library and sequencing

Sequencing library was generated with 1μg RNA sample using NEBNext UltraTM RNA Library Prep Kit for Illumina (NEB, USA) following manufacturer's recommendations. The index codes were added to attribute sequences to sample. The library was sequenced by Biomaker company (China). The clustering of the index-coded samples was performed on a cBot

Cluster Generation System using TruSeq PE Cluster Kit v4-cBot-HS (Illumina novaseq6000, PE150). The sequencing was processed on an Illumina platform and paired-end reads were generated. The clean data was obtained from the ploy-N reads by removing adapter and low quality reads [the ratio of N >10% and the ratio of the bases (Q≤10) > 50%]. The raw sequencing data was deposited into the NCBI sequence read archive (SRA) under the BioProject ID: PRJNA647242 (accession number: SRR12280784-SRR12280783).

## AS events identification and validation in the poplar leaves

Matrix clean reads were aligned to the *Populus trichocarpa* genome to define the gene reads locus [12], and the whole gene sequence were determined via NCBI (https://www.ncbi.nlm. nih.gov/assembly/GCF_000002775.3/) using the TopHat2 software. The assembled transcript isoforms were predicted by mapping to the corresponding gene model using Cufflinks program [13]. AS events in the poplar leaves were evaluated using ASprofile as previously described [14]. The expression level of genes or transcripts was evaluated via Fragments Per Kilobase of transcripts per Million fragments mapped (FPKM). FPKM = cDNA Fragments / [Mapped Fragments (Millions) * Transcript Length (kb)]. (cDNA Fragments means the fragments number aligned to a specific transcript. Mapped Fragments means the total fragments aligned to the transcripts).

Six AS genes in the poplar leaves having two AS isoforms and in size between 200-1000bp were selected for the validation of the sequencing results. The specific primers were designed at both ends of the gene (S1 Table). Each transcript fragment was amplified using cDNA template described above with the procedure of 95˚C for 10 min, followed by 35 cycles of 94˚C for 30 s, 55˚C for 30 s and 72˚C for 1 min. The PCR products were visualized using agarose gel electrophoresis.

## Function annotation of AS genes in the poplar leaves

The AS genes in the poplar leaves were aligned with DAVID database (http://david.abcc. ncifcrf.gov/) to obtain their functions, and classified as three clusters of molecular function, biological process and cell component according to GO (Gene Ontology) [15]. The differences between the genes and the background genes were evaluated by GOseq software. The gene enrichment analysis was conducted and defined by P value < 0.05. The pathway analysis of the AS genes was performed by KEGG (Kyoto Encyclopedia of Genes and Genomes) database. BH (Benjamini-Hochberg) method was used for P value adjustment. The pathway of P value < 0.05 was defined as significantly enriched.

## Statistic analysis

Based on the AS frequencies in the poplar leaves, the genes were classified into three groups: high AS genes (>5 AS events/gene), low AS genes (1–4 AS events/gene), and non-AS genes. Each group of the AS genes was then characterized according to their length, GC content, and expression levels. The data were analyzed statistically using Microsoft Excel 2013 for calculating mean and standard error. Duncan's multiple range test was used to compare the changes of the categories at the significant difference level of P <0.05.

# Results

## Overview of RNA-seq data

We obtained a total of 71.1 million clean data (7.11 Gb) by RNA-seq of the mature leaves of 84K poplar. The base Q30 percentage was 94.98% and the GC content was 44.93%. The

average read length is 150bp. The number of reads mapped to the reference genome accounted for 68.27% of the total clean reads; 65.94% were uniquely aligned and the remaining 2.33% mapped to multiple loci. Most of the multiple mapped loci contained rRNA, repeat-associated RNA, or part of tRNA sequences. The uniquely mapped reads were mainly mRNA and some tRNA and ncRNA sequences (S2 Table).

The majority of the mapped reads (93.52%) were in exonic regions, 2.84% were in intronic regions, and 3.64% were in nongene regions of the reference genome. The reads that were mapped to the exonic and intronic regions were used to analyze gene expression levels and to identify AS events for each gene.

## AS genes and events in the poplar leaves transcripts

In total, 28,200 genes were assembled and annotated using StringTie. Among the annotated genes, 13.51% of them underwent 9225 AS events in an average of 2.42 AS events per gene. Gene 1718, which encodes a glycerate kinase (GLYK), underwent 28 AS events, the highest number among all the annotated genes. GLYK is a key enzyme in glycolysis which also may affect DNA replication and repair, and stimulate viral RNA synthesis.

All the detected AS events in the poplar leaves were divided into four types, exon skipping (ES), intron retention (IR), alternative 5′ splicing site (A5SS), and alternative 3′ splicing site (A3SS). IR was the most common AS type, accounting for 43.86% (4046 events) of all AS events. IR was followed by A3SS and A5SS, which accounted for 23.75% and 23.71% of all AS events, respectively. ES accounted for only 8.68% of all AS events (Table 1). Some genes had more than one type of AS event; for example, gene Potri.002G124200.v3.0 (PABP1) underwent three types of AS events of IR, ES, and A5SS, and formed 6 varied AS isoforms in the poplar leaves (S1 Fig).

## Validation of AS events in the poplar leaves

We randomly selected six AS genes for validation by PCR. The results showed that all six genes each had two transcripts in the poplar leaves (Fig 1). The transcript sizes were consistent with the sizes obtained from the RNA-seq data, indicating that the sequencing data were reliable. For example, gene 6853 had two isoforms in the poplar leaves according to transcriptome data, one was from second intron retention and the other was the normal transcript. The amplified fragments were cloned and sequenced in the poplar leaves in size of 467bp and 411bp, respectively, and the results were consistent with expectation (S2 Fig).

**Table 1. Schematic diagram and distribution of the AS events in 84K poplar leaves.**

| Event type | AS pattern* | Events | Proportion (%) | Genes |
|---|---|---|---|---|
| IR | | 4046 | 43.86 | 2434 |
| A3SS | | 2191 | 23.75 | 869 |
| A5SS | | 2187 | 23.71 | 862 |
| ES | | 801 | 8.68 | 631 |
| Total | | 9225 | | 3810 |

*The line represents the intron, the black box represents the exon, the white box represents the exon in irregular splicing, and the gray box represents the retained intron; The straight line represents the normal splicing and the dotted line represents the irregular splicing.

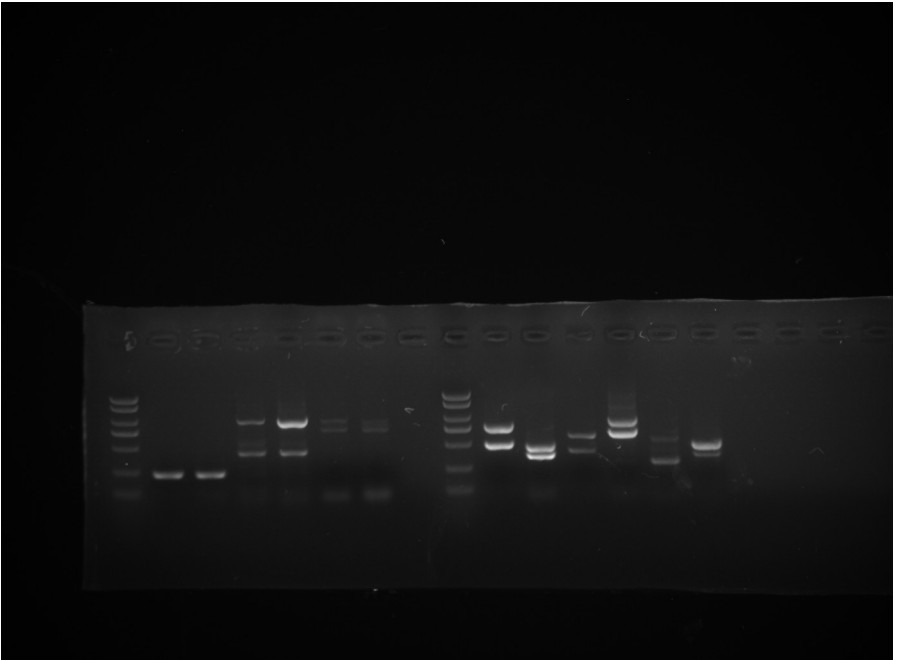

**Fig 1. Validation of AS events in the poplar leaves.** PCR results of six poplar AS genes by 1% (w/v) gel electrophoresis. 1: DL 2000 marker, 2: gene 1775, 3: gene 2340, 4: gene 14298, 5: gene 15649, 6: gene 5375, 7: gene 6835.

## Distribution of AS events on chromosomes

The AS events in the poplar leaves were mapped to 19 poplar chromosomes according to *P. trichocarpa* genome, but their distributions were uneven. Of them, chromosome 1 had the highest numbers of 1138 AS events (452 genes), and chromosome 19 had the smallest numbers of 198 AS events (90 genes) (Table 2). Moreover, chromosome 10 had the highest density of AS events (33.67 events/Mb) and AS genes (12.76%), while chromosome 11 and 19 had the smaller density of AS events (13.62 and 12.4233.67 events/Mb) and AS genes (5.99% and 6.19%). Even on a chromosome, AS events distributed unevenly such as some regions on chromosomes 8, 10, and 14 having more dense distributions of AS events (Fig 2).

## Characteristics of the AS Gene in the poplar leaves

Based on the AS frequencies, the genes in the poplar leaves were classified into three groups: high AS genes (>5 AS events/gene), low AS genes (1–4 AS events/gene), and non-AS genes. The RNA-seq results revealed there were 994 high AS genes, 2866 low AS genes, and 24,390 non-AS genes in the poplar transcriptome. The intron length, exon number, exon length, gene expression level, and GC content data were collected and evaluated among the three types of genes. Significantly, the AS events were positively correlated with the intron length, exon number, exon length, and gene expression level, but negatively correlated with GC content (Fig 3). Statistically (P<0.05), the average intron length of AS genes was 2.19 times that of non-AS genes, and the average intron length of high AS genes was 1.02 times that of low AS genes. The average exon number of AS genes was 2.35 times that of non-AS genes, and the average exon number of high AS genes was 1.22 times that of low AS genes. The average exon length of AS genes was 1.37 times that of non-AS genes, the average exon length of high AS genes was 1.06 times that of low AS genes. The average expression level of AS genes was 1.07 times that of

**Table 2. Distribution of the AS events and AS genes in the poplar leaves on poplar chromosomes.**

| Chromosome | AS events | Chromosome length (Mb) | AS density (AS events / Mb) | AS genes | Total gene number | AS density (AS genes / total genes) |
|---|---|---|---|---|---|---|
| 1 | 1138 | 50.49 | 22.54 | 452 | 4706 | 9.60% |
| 2 | 774 | 25.25 | 30.65 | 285 | 2589 | 11.01% |
| 3 | 543 | 21.79 | 24.92 | 229 | 2141 | 10.70% |
| 4 | 523 | 24.26 | 21.56 | 223 | 2334 | 9.55% |
| 5 | 634 | 25.89 | 24.49 | 276 | 2728 | 10.12% |
| 6 | 712 | 27.91 | 25.51 | 310 | 2771 | 11.19% |
| 7 | 363 | 15.61 | 23.25 | 148 | 1467 | 10.09% |
| 8 | 523 | 19.46 | 26.88 | 231 | 2267 | 10.19% |
| 9 | 343 | 12.95 | 26.49 | 156 | 1686 | 9.25% |
| 10 | 760 | 22.57 | 33.67 | 320 | 2507 | 12.76% |
| 11 | 252 | 18.5 | 13.62 | 102 | 1702 | 5.99% |
| 12 | 389 | 15.76 | 24.68 | 135 | 1452 | 9.30% |
| 13 | 352 | 16.32 | 21.57 | 155 | 1506 | 10.29% |
| 14 | 546 | 18.9 | 28.89 | 194 | 1929 | 10.06% |
| 15 | 323 | 15.28 | 21.14 | 145 | 1500 | 9.67% |
| 16 | 273 | 14.49 | 18.84 | 121 | 1432 | 8.45% |
| 17 | 280 | 16.08 | 17.41 | 112 | 1465 | 7.65% |
| 18 | 299 | 16.96 | 17.63 | 126 | 1385 | 9.10% |
| 19 | 198 | 15.94 | 12.42 | 90 | 1455 | 6.19% |

non-AS genes, and the average expression level of high AS genes was 1.12 times that of low AS genes. The average GC content of AS genes was 0.96 times that of non-AS genes, and the average GC content of high AS genes was 0.99 times that of low AS genes.

## Functional enrichment of AS genes in the poplar leaves

The Gene Ontology (GO) terms assigned to the AS genes in the poplar leaves were analyzed under the three main categories. Under the biological process, the 21 terms were related to metabolic processes, cellular processes, and single-organism processes (Fig 4). RNA processing and tRNA aminoacylation for protein translation etc were significantly enriched (S3 Table). Under cellular component, the 15 terms were related to cell part, cell, and organelle. Intracellular and Kinesin complex etc were more enriched. Under molecular function, the 15 terms

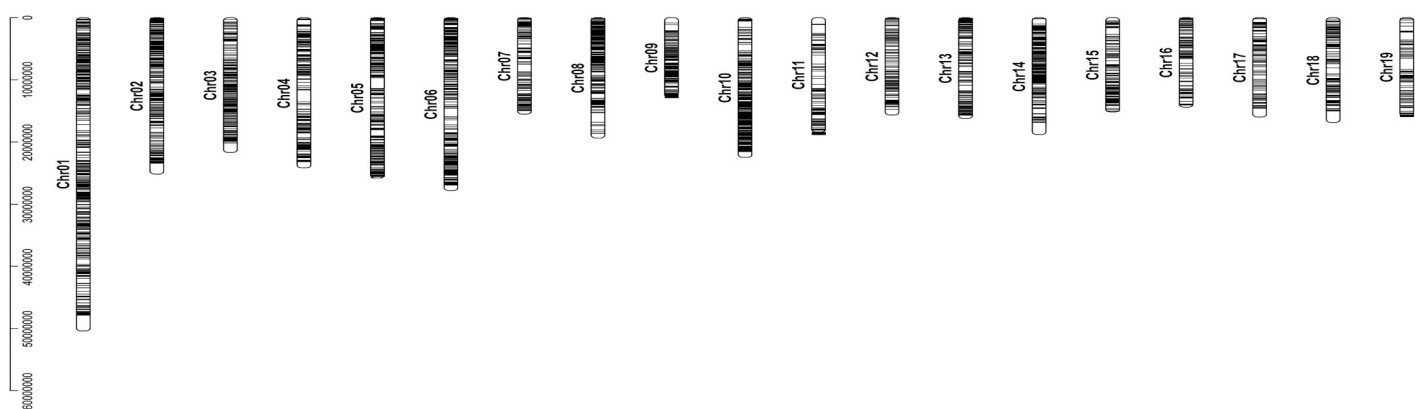

**Fig 2. Genome-wide distribution of the AS events in the poplar leaves on 19 poplar chromosomes.**

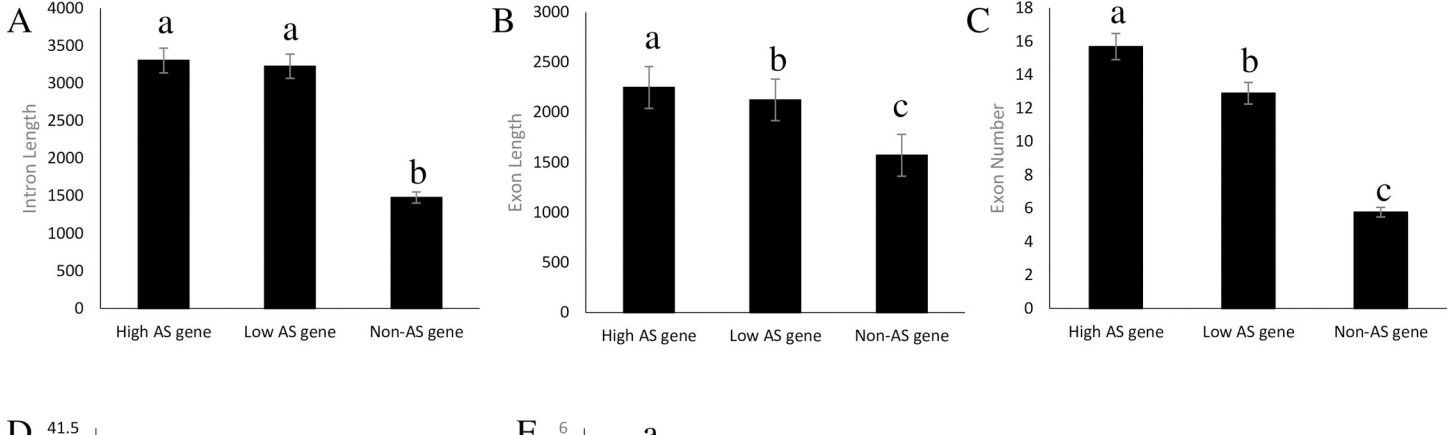

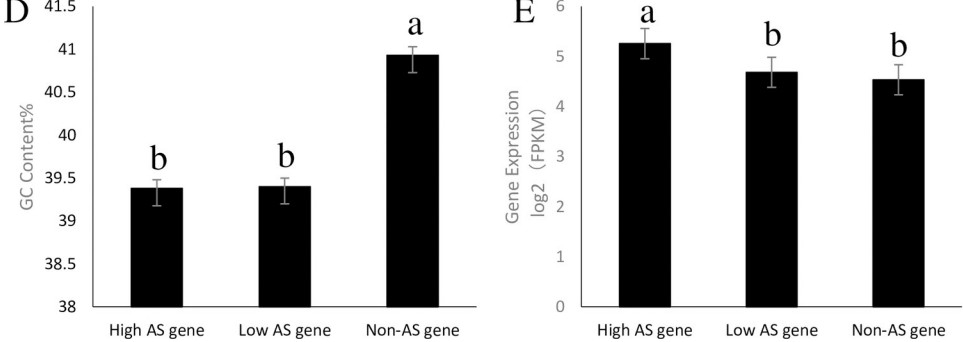

**Fig 3. Association between AS events and each factor in the poplar leaves.** All the transcribed genes in the poplar leaves were classified into three groups according to AS frequency in each gene as high AS genes (>5 AS events/gene), low AS genes (1–4 AS events/gene), and non-AS genes. A: Intron length, B: Exon length, C: Exon number, D: GC content, E: Expression level. The different letters indicate significant differences at the level of P < 0.05. The bars indicate SE.

were related to catalytic activity, binding, and transporter activity. Hydroquinone glucosyl-transferase activity and ATP binding etc were significantly enriched.

The KEGG analysis assigned 909 AS genes in the poplar leaves to 121 pathways (Fig 5). Among the significantly enriched pathways, seventy-four genes were in the spliceosome pathway, which is related to RNA processing and splicing (S3 Fig). Twenty-six genes were in the inositol phosphate metabolism pathway (S4 Fig) and Twenty-five genes were in the phosphatidylinositol signaling system pathway (S5 Fig), both of which are related to biosensory extracellular stimuli and signal transduction. Forty-one genes were in the mRNA surveillance pathway, which is related to clearance of invalid RNA in cells (Fig 6).

In comparison with the other AS reports, mRNA surveillance pathway was no doubt more concerned in the poplar leaves. Of them, PP2A (Potri.015G079300.v3.0) gene had three transcript isoforms, isoform1 was normal while isoform 2 and 3 separately had 1 and 3 exons deletion (Fig 7). Accordingly, it resulted in 38 and 88 amino acids missing at C side for isoform 2 and 3. The predicted secondary structure of the proteins showed that the abnormal isoforms had varied composition and distribution of alpha helix, extension chain, beta turn, and random coil in comparison of the normal isoform. These three transcripts individually had the FPKM value of 1.18, 2.56, and 6.07, and supposed to function differently.

## Discussion

### AS genes with species specificity

AS occurs widely in plant genomes and specifically by tissues. As the important organ, leaf contributes more for plant growth via respiration, photosynthesis, nutrient transformation,

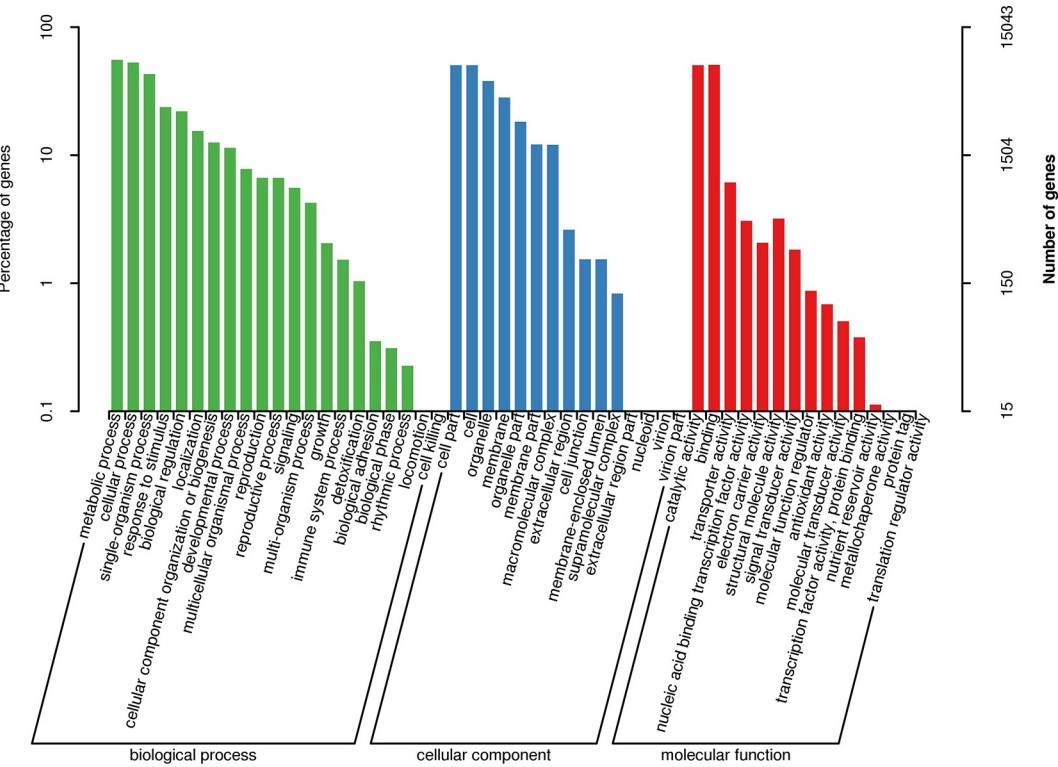

**Fig 4. Gene ontologies of AS genes in 84K poplar leaves.**

and transpiration. Thus, exploring the AS events in leaf could enhance our understanding for plant growth. In this study, we detected a total of 9225 AS events in the 84K poplar leaves transcriptome data. These AS events were from 3810 genes, which accounted for 13.51% of the

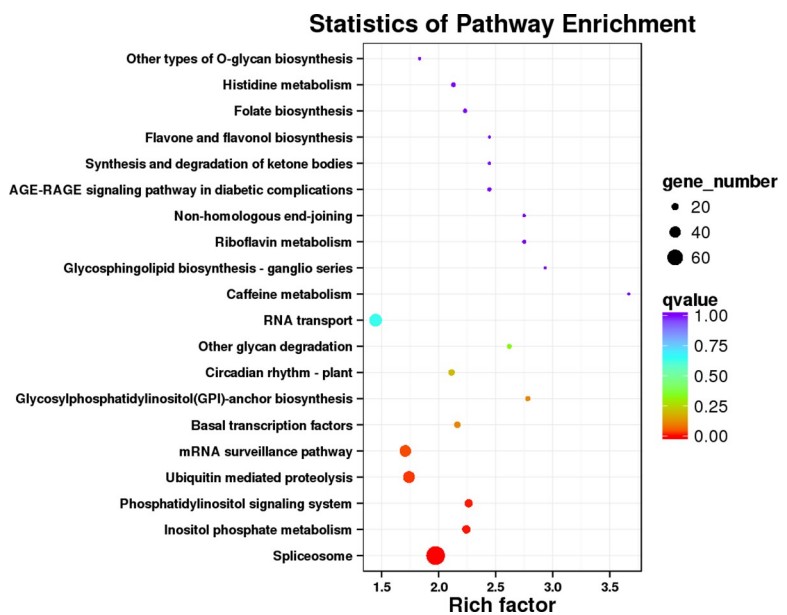

**Fig 5. KEGG enrichment of AS gene in 84K poplar leaves.**

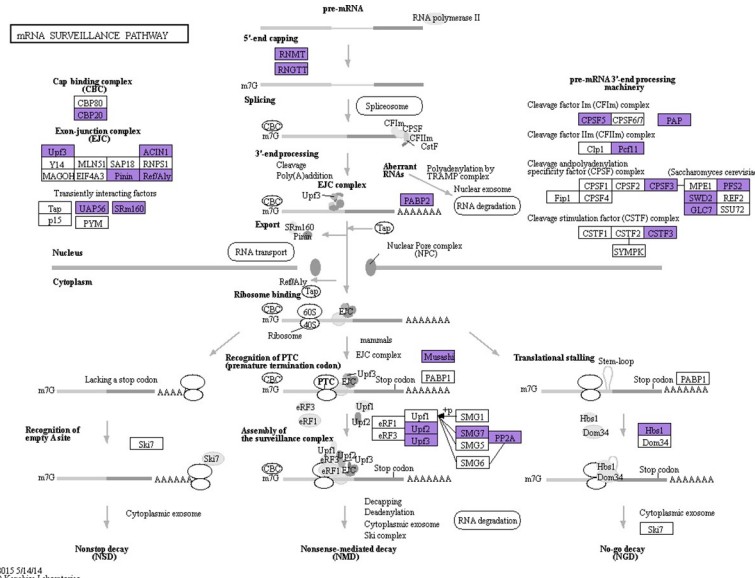

**Fig 6. The mRNA surveillance pathway and AS genes distribution in the poplar leaves.** The purple box represents AS genes.

genes in the transcriptome. Comparably, the AS events were detected in 8.73% of the transcripts in its parent of *Populus alba* var. *pyramidalis* [7]. Probably, the more isoforms by AS contribute more for the hybrid growth. In some other studies, the AS frequencies in kiwifruit [16], rice [17], and soybean [18] were 29%, 53.3%, and 63%, respectively, in the microorganisms *Magnaporthe grisea* [19], *Sphaerotheca fuliginea* [20], *Cryptococcus* [21], and *Aspergillus oryzae* [22] they were 1.6%, 3.6%, 4.2%, and 8.5%, respectively, and in *Drosophila melanogaster* [23] and human [24], there were 60% and 95%. Clearly, AS occurs more frequently in more advanced species, probably because genes in the advanced species have more introns and/or because of pleiotropism [25]. The high AS frequencies also explain why the advanced species have more complex behaviors and adapt well to environment variation [26], and why the gene numbers do not expand indefinitely as the species evolve [27]. Further, AS can be considered as a post-transcriptional regulation mechanism that acts as an effective strategy to mediate complex biological processes such as cell differentiation and organ development [28].

## Gene structure has significant effects on AS types

Among the four mechanisms (ES, IR, A5SS, A3SS) [29], ES is the most common in animals, whereas IR is more frequent in plants. For example, IR and ES account for 3% and 40% of all AS events in human [30], and 43.86% and 8.68% of all AS events in 84K poplar leaves, respectively. Similar situations were also revealed in plants of *Medicago truncatula* [31], *Populus trichocarpa* [32], *Arabidopsis* [33], *Oryza sativa* [17], and 15 animal species [34].

Gene structure is related to AS events. Association analysis of the gene composition elements between AS and non-AS genes in our study here showed that the occurrence of AS events in the poplar leaves was positively correlated with intron length, exon length, exon number, and expression level, and negatively correlated with GC content, which is consistent with previous studies in maize [35] and soybean [36]. It was even more distinguishly between plants and animals. For example, plant genes generally have smaller introns and slightly larger exons than human genes. The average exon length in rice [37], *Arabidopsis* [37], and human [38] is 254 bp, 217 bp, and 170 bp, respectively. While, approximately 50%–70% introns in

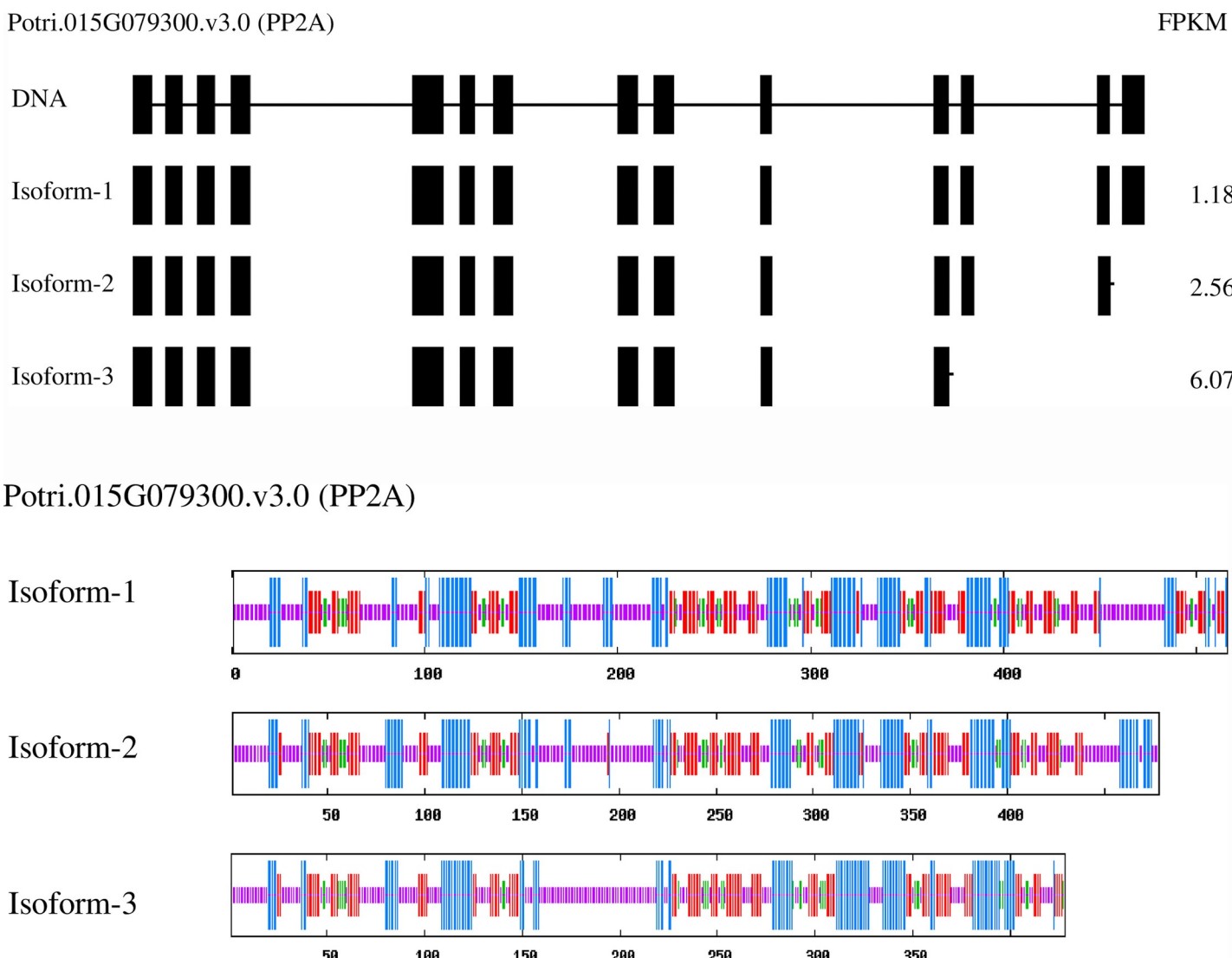

**Fig 7. The splicing structure of the PP2A transcripts isoforms in the poplar leaves.** A: mRNA composition of each transcript isoform (The block represents exon, line represents intron), B: the secondary structure of each transcript isoform (The blue parts represent alpha helix, the red parts represent extension chain, the green parts represent beta turn, and the purple parts represent random coil).

plant genes are ≤150 bp, whereas they average 5500 bp in human genes [39]. Thus, the differences in AS occurrences between plants and human described above can be understood if one considers that small introns are easier to retain and small exons are easier to skip.

## AS effects on poplar growth

The KEGG analysis of AS genes in the poplar leaves showed that several pathways were enriched, including the inositol phosphate metabolism (S4 Fig) and phosphatidylinositol signaling system pathways (S5 Fig). Inositol phosphate is known to play roles in signal transmission [40]. For example, IP3 is a secondary messenger that responds to calcium currents and participates in cell signal transduction [41]. IP6 participates in the DNA repair and mRNA translocation through synergistic or binding activation with corresponding protein factors [42]. IP1 actively responds to the abiotic stresses such as low temperature, drought, and salt

[43]. Therefore, the wide occurrence of AS events in inositol phosphate-related genes may provide new insights to understand the growth and development processes of poplar. Notably, the inositol related pathways have been less reported in plants such as *Camellia sinensis* [44] and wheat [45]. These results suggest that the AS events in inositol phosphate-related genes may be significant in poplar plant production.

Another enriched pathway that involved AS genes in the poplar leaves was the mRNA surveillance system (Fig 6), which could effectively remove non-functional RNA fragments caused by AS. In 84K poplar leaves, a total of 41 genes were found to be involved in this pathway, which may help to maintain its growth and development via the nonsense-mediated mRNA decay system, the nonstop mRNA decay system, and the no-go decay system. A well-organized system of this type was proposed previously for the AS process in soybean plants [36]. More studies are required to confirm the existence of such a system, and the function variation of the AS isoforms such as three transcript isoforms of PP2A gene.

Also, the AS evens in poplar here were evaluated in this paper by using the second generation sequencing technology. But the shortcomings in read length of the technique might cause the limitation and accuracy of the data although we increased the sequencing depth to minimize this impact. In future research, more validations could be conducted by PCR assay or full length isoform transcriptome analysis that could be complementary in read length and expression abundance.

## Conclusions

AS events in the poplar leaves were widely involved in metabolisms that would possibly affect poplar growth and development. AS occurrences were mostly in relation to the gene features such as intron length, exon number, gene expression level, and GC content. Further dissection of the AS isoforms function in the poplar leaves would be significant to understand and improve the mechanism of wooden plants production.

## Supporting information

**S1 Fig. Transcripts of the gene Potri.002G124200.v3.0(PABP1) in the poplar leaves.** (DOCX)

**S2 Fig. Transcripts alignment of two Illumina sequencing isoforms and two PCR fragments of the gene 6853 in the poplar leaves.** The bases in frame indicated PCR primer. The isoform1 was normal transcript and isoform2 was included an intron. (DOCX)

**S3 Fig. Splicesome pathway and AS genes distribution in the poplar leaves.** The purple box represents AS genes. (DOCX)

**S4 Fig. Inositol phosphate metabolism and AS genes distribution in the poplar leaves.** The purple box represents AS genes. (DOCX)

**S5 Fig. Phosphatidylinositol signaling system and AS genes distribution in the poplar leaves.** The purple box represents AS genes. (DOCX)

**S1 Table. The primer information for PCR validation.** (DOCX)

**S2 Table. Sequencing alignment results.**
(DOCX)

**S3 Table. The top 10 enriched genes in each GO category in the poplar leaves.**
(DOCX)

## Acknowledgments

We thank Liwen Bianji, Edanz Editing China (www.liwenbianji.cn/ac) for editing the English text of a draft of this manuscript.

## Author Contributions

**Investigation:** Ruixue Wang, Yang Ruixia.

**Methodology:** Ruixue Wang, Peng Yin, Yang Ruixia, Xiao Liu, Lie Luo.

**Project administration:** Jichen Xu.

**Resources:** Peng Yin.

**Software:** Lie Luo.

**Writing – original draft:** Ruixue Wang, Xiao Liu, Jichen Xu.

**Writing – review & editing:** Jichen Xu.

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
