## [Decision Letter · Decision Letter 0]

15 Jul 2020

PONE-D-20-17398

Genome-Wide Profiling of Alternative Splicing Genes in Populus

PLOS ONE

Dear Dr. Xu,

Thank you for submitting your manuscript to PLOS ONE. After careful consideration, we feel that it has merit but does not fully meet PLOS ONE’s publication criteria as it currently stands. Therefore, we invite you to submit a revised version of the manuscript that addresses the points raised during the review process.

As you will see, both reviewers are quite critical of your manuscript. I have decided to give you a chance to revise your work, but please take this very seriously. A revision consisting only of explaining away the reviewers comments would not be acceptable, you need to do major changes to the work and the writing.

We look forward to receiving your revised manuscript.

Kind regards,

Marc Robinson-Rechavi

Academic Editor

PLOS ONE

Journal Requirements:

2. We note that you are reporting an analysis of a microarray, next-generation sequencing, or deep sequencing data set. PLOS requires that authors comply with field-specific standards for preparation, recording, and deposition of data in repositories appropriate to their field. Please upload these data to a stable, public repository (such as ArrayExpress, Gene Expression Omnibus (GEO), DNA Data Bank of Japan (DDBJ), NCBI GenBank, NCBI Sequence Read Archive, or EMBL Nucleotide Sequence Database (ENA)). In your revised cover letter, please provide the relevant accession numbers that may be used to access these data. For a full list of recommended repositories, see http://journals.plos.org/plosone/s/data-availability#loc-omics or http://journals.plos.org/plosone/s/data-availability#loc-sequencing

'Funding

This work was supported by the National 267 Natural Science Foundation of China (#31672189) and Beijing Forestry University Undergraduate Training Program for Inovation and Entrepreneurship'

'The funders had no role in study design, data collection and analysis, decision to publish, or preparation of the manuscript.'

6. Please include a copy of Table 3 which you refer to in your text on page 6.

7. Please include captions for your Supporting Information files at the end of your manuscript, and update any in-text citations to match accordingly. Please see our Supporting Information guidelines for more information: http://journals.plos.org/plosone/s/supporting-information

Reviewers' comments:

Reviewer's Responses to Questions

**Comments to the Author**

1. Is the manuscript technically sound, and do the data support the conclusions?

Reviewer #1: Partly

Reviewer #2: Partly

2. Has the statistical analysis been performed appropriately and rigorously? 

Reviewer #1: No

Reviewer #2: Yes

3. Have the authors made all data underlying the findings in their manuscript fully available?

Reviewer #1: No

Reviewer #2: Yes

4. Is the manuscript presented in an intelligible fashion and written in standard English?

Reviewer #1: No

Reviewer #2: No

5. Review Comments to the Author

Reviewer #1: The authors applied Illumina Next-Generation Sequencing (NGS) technology to profile the alternative splicing (AS) events in the leaf transcriptome of 2-month old Populus 84K, which has not being reported for this specific poplar hybrid. While the results could be useful for future research, there are some shortcomings in this study, making the experimental design and findings to be very preliminary. A major revision is necessary focusing on the shortcomings and comments raised appended below. Thus, I would not recommend it for publication in PloS One in its present form.

Shortcomings:

1. The introduction of third-generation long-read sequencing technology (e.g. Pacific Biosciences) has overcome the traditional issues on NGS technology such as bias, false positive and misassembly. While many published research has been using long-read technology to report plant AS characterisation (including Populus species), this study presented the NGS-based findings. To improve the accuracy and impact of the results, the authors should apply long-read technology to make the ‘reference’ isoform library and use NGS data for error correction and expression counting. Alternatively, in the current manuscript, the authors should at least discuss this shortcoming and make clear to the readers about the limitation and accuracy of the data/results.

2. This study lacks a very clear and convincing problem statement and research objective. If the research is aimed to first characterise the unknown AS events in the Poplar hybrid, the samples used should cover all major plant tissues such as root, xylem, phloem etc. and not limited to the only leaf with very low transcript/isoform coverage. AS events have been known to be tissue, environment and individual specific. Also, the replication of the sample in this study seems to be only 1. With only leaf sample being sampled, the authors should justify the significance of using the leaf sample, e.g. specific scientific or economic importance of Poplar leaf, and redirect the findings and discussion toward this angle/direction.

3. There are a number of publications on the AS characterisation from Populus species, including the Populus alba, the parent of this 84K hybrid. This reviewer does not understand why there is no introduction about these scientific reports in the introduction and comparison in the Results/Discussion session.

4. Additional analysis, such as alternative polyadenylation (APA) would add more values and impact to this research and paper.

5. The Materials and Methods section is incomplete with the missing experimental procedure such as expression count, cloning and sequencing, statistical analysis. The current description on the methodology is also not sufficient, clear or detail for reproducibility.

6. The completeness of the transcriptome made in this study is unknown and should be evaluated with analysis such as BUSCO. With only one plant tissue (i.e. leaf) is presented, we have to know how comprehensive or completeness of the leaf transcriptome presented, which will provide a hint on the potential capturing of “all” AS genes. Also, the authors should consider adding GO enrichment analysis and qPCR validation of FPKM expression values.

Comments:

1. Lines 1-2: the title has to be more specific (e.g. species, leaf) due to the shortcomings listed in Shortcoming No. 2 mentioned above.

2. Lines 17-20: include percentage for all types of AS events and AS density for the chromosomes discussed.

3. Line 24 and Lines 258-262: a conclusion is missing or weakly summarised, mainly because this paper has no direction as in Shortcoming No. 3 mentioned above. Perhaps the authors can elaborate on how the findings should help to shed light on plant development or wooden plant production.

4. Lines 29-49: Introduction on fundamental information or knowledge (e.g. DNA-RNA transcription, Intron/AS frequency in plants) is not adequate for general readers, including those new to this field/area. The arrangement has to be improved, e.g. the sentence from Lines 34-35 should move forward and place after sentence from lines 30-31.

5. Lines 50-51: expand the literature specific to plants (e.g. angiosperms, gymnosperms, spermatophytes, pteridophytes, lycophytes etc.) and poplar species.

6. Line 53: when “strong resistance” is mentioned, what does it mean? Abiotic? Biotic? Environment?

7. Line 56: add problem statement, objective, justification and rationale of this study by referring to Shortcoming No. 2 mentioned above.

8. Line 62: replication?

9. Line 70: data repository (raw sequencing file) to the public database such as NCBI or ENA?

10. Line 74: define clearly the company and country information

11. Line 76: define clearly which Illumina platform

12. Line 77: what is the read length?

13. Line 78: include quality assessment criteria. Q30?

14. Line 80: cite the genome paper and source (i.e. repository database where the sequences are obtained)

15. Line 80: “.. and the gene…”: the gene is referred to the Populus trichocarpa genes presented in the genome?

16. Line 83: how the transcriptome assembly being done? Software and methodology.

17. Line 85: how the FPKM value being counted and calculated? The methodology is missing.

18. Line 86: any selection criteria for the sequencing results validation? E.g. gene size, frequency of introns, AS pattern type etc. It cannot be too “random”, or else it could be bias

19. Line 90: include materials and methods on “Cloning and Sequencing” for PCR validation

20. Lines 92-95: include reference/website link for databases such as GO and DAVID

21. Lines 94-95: use the term “enrichment analysis”

22. Line 99: statistical analysis is missing (software, type of test and methodology)

23. Table 1: check the typo and alignment. Is the “clean bases” equivalent to “total GB of data obtained”?

24. Lines 128-129: when “some genes had more than one type of AS event” is mentioned, more information or details should be provided or elaborated.

25. Lines 116-129: cite the Table/Figure when describing the number/results, same for remaining sections of result description.

26. Table 2: consider to add a “Total” row to show the sum values

27. Lines 139: give the identifies of the genes when “... all five genes…” are mentioned

28. Lines 141-144: instead of showing specific example without much impact, we suggest to discuss or illustrate more solid data on the validation, e.g. comparison of sizes from NGS and size from PCR/cloning

29. Line 146: indicate the number of chromosomes

30. Lines 146-147: do you mean equal (even) or unequal (uneven) distribution?

31. Lines 151-152: how you judge on this? Visually, this reviewer see chromosomes 1, 5 and 6 have a comparable high density of AS as well

32. Lines 151-152: would it be useful to provide the average AS events/Mb (one mean value) for all chromosomes?

33. Lines 161-170: when mentioning about “XX times higher/lower than” it should be supported by statistics (e.g. P<0.05) and described in the text

34. Lines 173: did the authors verify some of the FPKM using real-time quantitative PCR? Or else, can the FPKM results be trusted?

35. Lines 171-176: please justify the rationale to show/describe this. Or else, it is better to remove it and replace with more useful information, e.g. a list of important genes exclusively expressed (with specific biochemistry importance and interest) in the leaf with unique and interesting splicing events. It will be good to verify the AS pattern and FPKM values of this set of “genes of interest” using cloning and qPCR, respectively.

36. Lines 178-183: without the GO enrichment analysis, the results presented are weak and less meaningful

37. Lines 186-289: do not start a sentence with a number

38. Line 192: why only this pathway being selected? Any justification?

39. Lines 204-209: it will be more informative to compare the AS pattern among Poplar species as several AS papers on Poplar are available

40. Line 211: “higher species” means multicellular form? Please be specific.

41. Lines 216-271: please add citation(s)

42. Lines 221: without only one example for each group (i.e. human and Populus 84k), it is less convincing to represent the entire animal and plant groups, respectively.

43. Lines 227-233: focus more on the comparison within plant groups (with more examples)

44. Line 245: “these pathways” refers to?

45. Lines 259-261: the sentences have to be rephrased and improved

46. Figure legends: the titles and descriptions are too brief. For example, revised them to “1% (w/v) gel electrophoresis of the… 1, DNA marker, ….” for Figure 1, “Genome-wide distribution of the … 19 poplar chromosomes” for Figure 2, “The splicing structure of …” for Figure 6.

47. Supplementary Materials: The authors should consider to include Fig. S1-4 in the main document (or some of them) to provide more depth in the discussion of the results. If the number of figures is too many, maybe can move Fig.1 and Fig. 4 to the supplementary materials.

Reviewer #2: The study submitted by Wang et al. is focused on the raw identification of AS events in Populus 84K. The data obtained are interesting and will be usefull in future studies, but the paper need to be reviewed by a English mothertongue and the "Introduction"and "Discussion" text need to be reformulate because it is not fluent.

This paper can be accepted with major revisions.

See below both major and minor revisions.

Please change the genus and species names in italic, in the whole manuscript.

Title

I suggest to add “spp.” after Populus

Abstract

The abstract is written very well. Please check the the word “Populus” in always in italic. Just few suggestions.

Lines 10-11: Change the step in “Alternative splicing (AS) is a post-transcriptional process common in higher plants and essential for regulation of environmental fitness of plants.”

Line 11: change the step in “In the present study, we dissected the poplar alternative splicing events in order to understand their effects on plant growth and development.”

Keywords:

Please do not use the same words you have in the title. For example, change “Populus” in “Poplar” and eliminate Alternative splicing

Introduction

The introduction is too short and not structured very well. You report a list of genes known to have isoforms that functionally influence plant stress response as a shopping list. Please reformulate lines 34-49. And add more informations about omic studies carried out on Populus 84k or other poplar genotypes.

Lines 31-33: the AS events do not occur in plants only for adaptation to abiotic conditions, so please change this part adding even the biotic response of plants, see and add to the references “Zheng, Z., Appiano, M., Pavan, S., Bracuto, V., Ricciardi, L., Visser, R. G., ... & Bai, Y. (2016). Genome-wide study of the tomato SlMLO gene family and its functional characterization in response to the powdery mildew fungus Oidium neolycopersici. Frontiers in plant science, 7, 380.” and “De Palma, M., Salzano, M., Villano, C., Aversano, R., Lorito, M., Ruocco, M., ... & Tucci, M. (2019). Transcriptome reprogramming, epigenetic modifications and alternative splicing orchestrate the tomato root response to the beneficial fungus Trichoderma harzianum. Horticulture research, 6(1), 1-15.”

Lines 34-38: first you have to start the step concerning CCA1 isoforms saying that it is an example. Second, the paper you cited is not the one about the study of CCA1 isoforms, please change the citation and use Seo et al., Plant Cell. 2012, 24: 2427-2442. 10.1105/tpc.112.098723. Third, lines 35-37 are the same of Kwon et al, it is plagiarism! And lines 37-38 are not correct, please reformulate correctly!

Materials and methods

Please provide information about biological and technical replicates

Some details are missing. Add for each product used the company and the location (e.g. line 64 Aidlab, line 65 Takara…).

Line 64: change “trizol” in “Trizol”

Line 67: eliminate “Nanodrop” in brackets

Line 99: change in “significantly”

Results

The results are very interesting but not very well written. There is an abuse of the verb “had undergone” please change the text to avoid this term.

Change Table 2 in Figure 1 and modify the other figures accordingly

Table 1 legend is not complete!

Line 141: which gene are you talking about? In the figure 1 legend it is named 6835, in the text 6853.

Line 144: add a Supplementary figure with sequencing results, for example an alignment between cloned fragments and Illumina results.

Figure 1: change the legend in “Validation of AS events. PCR results of six genes randomly selected. 1: marker (ADD THE MARKER TYPE!), 2: gene 1775, 3: gene 2340, 4: gene 14298, 5: gene 15649, 6: gene 5375, 7: gene 6835.

Line 146: add information about the genome used

Line 196: change “isoform” in “isoform”

Line 198: check the font

Discussion

Line 212: add a reference

Line 219: move the citations at the end of the step.

6. PLOS authors have the option to publish the peer review history of their article (what does this mean?). If published, this will include your full peer review and any attached files.

Reviewer #1: No

Reviewer #2: No

---

## [Author Response · Author response to Decision Letter 0]

2 Sep 2020

Journal Requirements:

Reply: We ensure that your manuscript meets PLOS ONE's style requirements

2. We note that you are reporting an analysis of a microarray, next-generation sequencing, or deep sequencing data set. PLOS requires that authors comply with field-specific standards for preparation, recording, and deposition of data in repositories appropriate to their field. Please upload these data to a stable, public repository (such as ArrayExpress, Gene Expression Omnibus (GEO), DNA Data Bank of Japan (DDBJ), NCBI GenBank, NCBI Sequence Read Archive, or EMBL Nucleotide Sequence Database (ENA)). In your revised cover letter, please provide the relevant accession numbers that may be used to access these data. For a full list of recommended repositories, 

 Reply: We deposited our transcription data in NCBI GenBank and mentioned in the manuscript and cover letter.

'Funding

This work was supported by the National 267 Natural Science Foundation of China (#31672189) and Beijing Forestry University Undergraduate Training Program for Inovation and Entrepreneurship'

'The funders had no role in study design, data collection and analysis, decision to publish, or preparation of the manuscript.'

 Reply: The funding information was described in cover letter

4. PLOS ONE now requires that authors provide the original uncropped and unadjusted images underlying all blot or gel results reported in a submission’s figures or Supporting Information files. When you submit your revised manuscript, please ensure that your figures adhere fully to these guidelines and provide the original underlying images for all blot or gel data reported in your submission. 

In your cover letter, please note whether your blot/gel image data are in Supporting Information or posted at a public data repository, provide the repository URL if relevant, and provide specific details as to which raw blot/gel images, if any, are not available. 

 Reply: We ensure that the figures adhere fully to the guidelines 

 Reply: It was done

6. Please include a copy of Table 3 which you refer to in your text on page 6.

 Reply: It was done

7. Please include captions for your Supporting Information files at the end of your manuscript, and update any in-text citations to match accordingly. 

Reply: It was done

Reviewers' comments:

Reviewer's Responses to Questions

Comments to the Author

1. Is the manuscript technically sound, and do the data support the conclusions?

Reviewer #1: Partly

Reviewer #2: Partly

 Reply: We modified the conclusions

2. Has the statistical analysis been performed appropriately and rigorously?

Reviewer #1: No

Reviewer #2: Yes

 Reply: We modified the related part

3. Have the authors made all data underlying the findings in their manuscript fully available?

Reviewer #1: No

Reviewer #2: Yes

Reply: We modified the related part

4. Is the manuscript presented in an intelligible fashion and written in standard English?

Reviewer #1: No

Reviewer #2: No

Reply: We modified the writing

5. Review Comments to the Author

 Reply: It was done

Reviewer #1: 

1. The introduction of third-generation long-read sequencing technology (e.g. Pacific Biosciences) has overcome the traditional issues on NGS technology such as bias, false positive and misassembly. While many published research has been using long-read technology to report plant AS characterisation (including Populus species), this study presented the NGS-based findings. To improve the accuracy and impact of the results, the authors should apply long-read technology to make the ‘reference’ isoform library and use NGS data for error correction and expression counting. Alternatively, in the current manuscript, the authors should at least discuss this shortcoming and make clear to the readers about the limitation and accuracy of the data/results.

 Reply: I agree with the reviewer’s points. Definitely, NGS technology has some shortcomings such as reading length in comparison with the whole length transcriptome technology but it also has the advancement in expression abundance. We did modification in discussion about the limitation and accuracy of NGS technology in AS characterisation.

2. This study lacks a very clear and convincing problem statement and research objective. If the research is aimed to first characterise the unknown AS events in the Poplar hybrid, the samples used should cover all major plant tissues such as root, xylem, phloem etc. and not limited to the only leaf with very low transcript/isoform coverage. AS events have been known to be tissue, environment and individual specific. Also, the replication of the sample in this study seems to be only 1. With only leaf sample being sampled, the authors should justify the significance of using the leaf sample, e.g. specific scientific or economic importance of Poplar leaf, and redirect the findings and discussion toward this angle/direction. 

Reply: AS events were much complicated and specifically by tissues. Our study here just focused on leaf and hopefully to provide some initial opinions of its effect on plant growth. Thus we modified the description of our objective. 

 3. There are a number of publications on the AS characterization from Populus species, including the Populus alba, the parent of this 84K hybrid. This reviewer does not understand why there is no introduction about these scientific reports in the introduction and comparison in the Results/Discussion session.

 Reply: We add some more publications of populus in introduction and discussion sessions.

4. Additional analysis, such as alternative polyadenylation (APA) would add more values and impact to this research and paper.

 Reply: We agree with the reviewer’s point. The manuscript here is an initial report. We hope to conduct the AS mechanism analysis including APA in next.

5. The Materials and Methods section is incomplete with the missing experimental procedure such as expression count, cloning and sequencing, statistical analysis. The current description on the methodology is also not sufficient, clear or detail for reproducibility.

 Reply: We modified the Materials and Methods section in details

6. The completeness of the transcriptome made in this study is unknown and should be evaluated with analysis such as BUSCO. With only one plant tissue (i.e. leaf) is presented, we have to know how comprehensive or completeness of the leaf transcriptome presented, which will provide a hint on the potential capturing of “all” AS genes. Also, the authors should consider adding GO enrichment analysis and qPCR validation of FPKM expression values.

 Reply: For the transcription data evaluation, some modifications were added in the methods and results sessions. 

1. Lines 1-2: the title has to be more specific (e.g. species, leaf) due to the shortcomings listed in Shortcoming No. 2 mentioned above.

 Reply: It was done

2. Lines 17-20: include percentage for all types of AS events and AS density for the chromosomes discussed.

 Reply: It was done

3. Line 24 and Lines 258-262: a conclusion is missing or weakly summarised, mainly because this paper has no direction as in Shortcoming No. 3 mentioned above. Perhaps the authors can elaborate on how the findings should help to shed light on plant development or wooden plant production.

 Reply: It was done

4. Lines 29-49: Introduction on fundamental information or knowledge (e.g. DNA-RNA transcription, Intron/AS frequency in plants) is not adequate for general readers, including those new to this field/area. The arrangement has to be improved, e.g. the sentence from Lines 34-35 should move forward and place after sentence from lines 30-31.

 Reply: The first paragraph described the general process of the mRNA formation. And the second paragraph turned to the changed cases of AS. I do think this order might be helpful for the new to follow it. Therefore, I just changed a little.

5. Lines 50-51: expand the literature specific to plants (e.g. angiosperms, gymnosperms, spermatophytes, pteridophytes, lycophytes etc.) and poplar species.

 Reply: Several publications were added.

6. Line 53: when “strong resistance” is mentioned, what does it mean? Abiotic? Biotic? Environment?

 Reply: It was modified.

7. Line 56: add problem statement, objective, justification and rationale of this study by referring to Shortcoming No. 2 mentioned above. 

Reply: It was modified.

8. Line 62: replication?

 Reply: It was modified.

9. Line 70: data repository (raw sequencing file) to the public database such as NCBI or ENA?

 Reply: It was modified.

10. Line 74: define clearly the company and country information

 Reply: It was modified.

11. Line 76: define clearly which Illumina platform

 Reply: It was done.

12. Line 77: what is the read length?

Reply: It was in results session

13. Line 78: include quality assessment criteria. Q30?

 Reply: It was in results session

14. Line 80: cite the genome paper and source (i.e. repository database where the sequences are obtained)

Reply: It was modified.

15. Line 80: “.. and the gene…”: the gene is referred to the Populus trichocarpa genes presented in the genome?

 Reply: Both populus species of 84K and Populus trichocarpa have similar genome composition. Thus, we used the sequenced genome of Populus trichocarpa as the reference to define the gene reads locus, and further determine the whole gene sequence via NCBI blast. Some modifications were done. 

16. Line 83: how the transcriptome assembly being done? Software and methodology.

 Reply: Matrix clean reads were aligned to the Populus trichocarpa genome to define the gene reads locus.

17. Line 85: how the FPKM value being counted and calculated? The methodology is missing.

 Reply: It was completed.

18. Line 86: any selection criteria for the sequencing results validation? E.g. gene size, frequency of introns, AS pattern type etc. It cannot be too “random”, or else it could be bias

 Reply: It was modified

19. Line 90: include materials and methods on “Cloning and Sequencing” for PCR validation

 Reply: It was modified

20. Lines 92-95: include reference/website link for databases such as GO and DAVID

 Reply: It was done

21. Lines 94-95: use the term “enrichment analysis”

 Reply: It was done

22. Line 99: statistical analysis is missing (software, type of test and methodology)

 Reply: It was done

23. Table 1: check the typo and alignment. Is the “clean bases” equivalent to “total GB of data obtained”?

 Reply: It was modified

24. Lines 128-129: when “some genes had more than one type of AS event” is mentioned, more information or details should be provided or elaborated.

 Reply: It was modified

25. Lines 116-129: cite the Table/Figure when describing the number/results, same for remaining sections of result description.

 Reply: It was modified

26. Table 2: consider to add a “Total” row to show the sum values

Reply: It was modified

27. Lines 139: give the identifies of the genes when “... all five genes…” are mentioned

Reply: It was modified

28. Lines 141-144: instead of showing specific example without much impact, we suggest to discuss or illustrate more solid data on the validation, e.g. comparison of sizes from NGS and size from PCR/cloning

 Reply: It was modified

29. Line 146: indicate the number of chromosomes

 Reply: It was modified

30. Lines 146-147: do you mean equal (even) or unequal (uneven) distribution?

 Reply: It was modified

31. Lines 151-152: how you judge on this? Visually, this reviewer see chromosomes 1, 5 and 6 have a comparable high density of AS as well

 Reply: It was modified

32. Lines 151-152: would it be useful to provide the average AS events/Mb (one mean value) for all chromosomes?

 Reply: It showed in table2

33. Lines 161-170: when mentioning about “XX times higher/lower than” it should be supported by statistics (e.g. P<0.05) and described in the text

 Reply: It was modified

34. Lines 173: did the authors verify some of the FPKM using real-time quantitative PCR? Or else, can the FPKM results be trusted?

 Reply: We had test 6 genes as shown in Fig1. The results showed the tendency of the isoforms FPKM was identical with the transcriptome data. Thus we verify that the description here can be trusted.

35. Lines 171-176: please justify the rationale to show/describe this. Or else, it is better to remove it and replace with more useful information, e.g. a list of important genes exclusively expressed (with specific biochemistry importance and interest) in the leaf with unique and interesting splicing events. It will be good to verify the AS pattern and FPKM values of this set of “genes of interest” using cloning and qPCR, respectively.

 Reply: These descriptions were removed.

36. Lines 178-183: without the GO enrichment analysis, the results presented are weak and less meaningful

Reply: It was modified

37. Lines 186-289: do not start a sentence with a number

Reply: It was modified

38. Line 192: why only this pathway being selected? Any justification?

 Reply: In the other AS studies, mRNA surveillance pathway was less reported. AS events probably caused more invalid RNAs that might influence the regular cell activity. Thus, mRNA surveillance pathway is significant for plant growth. We did some modification here and further explained in discussion session. 

39. Lines 204-209: it will be more informative to compare the AS pattern among Poplar species as several AS papers on Poplar are available

 Reply: It was done

40. Line 211: “higher species” means multicellular form? Please be specific.

 Reply: It was modified

41. Lines 216-271: please add citation(s)

 Reply: It was modified

42. Lines 221: without only one example for each group (i.e. human and Populus 84k), it is less convincing to represent the entire animal and plant groups, respectively.

 Reply: It was modified

43. Lines 227-233: focus more on the comparison within plant groups (with more examples)

 Reply: For plants, we placed the populus, maize and soybean expamples there. We also think that the comparison between plants and human could also enance the point. Thus, some modifications were done

44. Line 245: “these pathways” refers to?

 Reply: It was modified

45. Lines 259-261: the sentences have to be rephrased and improved

 Reply: It was modified

46. Figure legends: the titles and descriptions are too brief. For example, revised them to “1% (w/v) gel electrophoresis of the… 1, DNA marker, ….” for Figure 1, “Genome-wide distribution of the … 19 poplar chromosomes” for Figure 2, “The splicing structure of …” for Figure 6.

 Reply: It was modified

47. Supplementary Materials: The authors should consider to include Fig. S1-4 in the main document (or some of them) to provide more depth in the discussion of the results. If the number of figures is too many, maybe can move Fig.1 and Fig. 4 to the supplementary materials.

 Reply: It was modified

Reviewer #2: The study submitted by Wang et al. is focused on the raw identification of AS events in Populus 84K. The data obtained are interesting and will be usefull in future studies, but the paper need to be reviewed by a English mothertongue and the "Introduction"and "Discussion" text need to be reformulate because it is not fluent.

 Reply: The manuscript was reviewed by Liwen Bianji, Edanz Editing China (www.liwenbianji.cn/ac) for editing the English text. 

Please change the genus and species names in italic, in the whole manuscript.

 Reply: It was modified

Title

I suggest to add “spp.” after Populus

 Reply: The populus here is just a hybrid variety. It was modified

Abstract

The abstract is written very well. Please check the the word “Populus” in always in italic. Just few suggestions.

 Reply: It was modified

Lines 10-11: Change the step in “Alternative splicing (AS) is a post-transcriptional process common in higher plants and essential for regulation of environmental fitness of plants.”

 Reply: It was modified

Line 11: change the step in “In the present study, we dissected the poplar alternative splicing events in order to understand their effects on plant growth and development.”

Reply: It was modified

Keywords:

Please do not use the same words you have in the title. For example, change “Populus” in “Poplar” and eliminate Alternative splicing

Reply: It was modified

Introduction

The introduction is too short and not structured very well. You report a list of genes known to have isoforms that functionally influence plant stress response as a shopping list. Please reformulate lines 34-49. And add more informations about omic studies carried out on Populus 84k or other poplar genotypes.

Reply: It was modified

Lines 31-33: the AS events do not occur in plants only for adaptation to abiotic conditions, so please change this part adding even the biotic response of plants, see and add to the references “Zheng, Z., Appiano, M., Pavan, S., Bracuto, V., Ricciardi, L., Visser, R. G., ... & Bai, Y. (2016). Genome-wide study of the tomato SlMLO gene family and its functional characterization in response to the powdery mildew fungus Oidium neolycopersici. Frontiers in plant science, 7, 380.” and “De Palma, M., Salzano, M., Villano, C., Aversano, R., Lorito, M., Ruocco, M., ... & Tucci, M. (2019). Transcriptome reprogramming, epigenetic modifications and alternative splicing orchestrate the tomato root response to the beneficial fungus Trichoderma harzianum. Horticulture research, 6(1), 1-15.”

Reply: It was modified

Lines 34-38: first you have to start the step concerning CCA1 isoforms saying that it is an example. Second, the paper you cited is not the one about the study of CCA1 isoforms, please change the citation and use Seo et al., Plant Cell. 2012, 24: 2427-2442. 10.1105/tpc.112.098723. Third, lines 35-37 are the same of Kwon et al, it is plagiarism! And lines 37-38 are not correct, please reformulate correctly!

Reply: All were modified

Materials and methods

Please provide information about biological and technical replicates

Reply: It was modified

Some details are missing. Add for each product used the company and the location (e.g. line 64 Aidlab, line 65 Takara…).

Reply: It was modified

Line 64: change “trizol” in “Trizol”

Reply: It was modified

Line 67: eliminate “Nanodrop” in brackets

Reply: It was modified

Line 99: change in “significantly”

Reply: It was modified

Results

The results are very interesting but not very well written. There is an abuse of the verb “had undergone” please change the text to avoid this term.

Reply: It was modified

Change Table 2 in Figure 1 and modify the other figures accordingly

Reply: It was modified

Table 1 legend is not complete!

Reply: It was modified

Line 141: which gene are you talking about? In the figure 1 legend it is named 6835, in the text 6853.

Reply: It was modified

Line 144: add a Supplementary figure with sequencing results, for example an alignment between cloned fragments and Illumina results.

Reply: It was modified

Figure 1: change the legend in “Validation of AS events. PCR results of six genes randomly selected. 1: marker (ADD THE MARKER TYPE!), 2: gene 1775, 3: gene 2340, 4: gene 14298, 5: gene 15649, 6: gene 5375, 7: gene 6835.

Reply: It was modified

Line 146: add information about the genome used 

Reply: It was modified

Line 196: change “isoform” in “isoform”

Reply: It was done

Line 198: check the font

Reply: It was done

Discussion

Line 212: add a reference

Reply: It was done

Line 219: move the citations at the end of the step.

 Reply: It was done

---

## [Decision Letter · Decision Letter 1]

6 Oct 2020

PONE-D-20-17398R1

Genome-Wide Profiling of Alternative Splicing Genes in Hybrid Poplar (P.alba×P.glandulosa cv.84K) Leaf

PLOS ONE

Dear Dr. Xu,

Thank you for submitting your manuscript to PLOS ONE. After careful consideration, we feel that it has merit but does not fully meet PLOS ONE’s publication criteria as it currently stands. Therefore, we invite you to submit a revised version of the manuscript that addresses the points raised during the review process.

Please address the minor comments of Reviewer 1. In addition, please check the writing carefully. A rapid reading finds many small mistakes, such as "Chromosomes 10" (should be "Chromosome 10") in the Abstract, or "Of thm" (should be "Of them") on line 182.

We look forward to receiving your revised manuscript.

Kind regards,

Marc Robinson-Rechavi

Academic Editor

PLOS ONE

Reviewers' comments:

Reviewer's Responses to Questions

**Comments to the Author**

1. If the authors have adequately addressed your comments raised in a previous round of review and you feel that this manuscript is now acceptable for publication, you may indicate that here to bypass the “Comments to the Author” section, enter your conflict of interest statement in the “Confidential to Editor” section, and submit your "Accept" recommendation.

Reviewer #1: All comments have been addressed

Reviewer #2: All comments have been addressed

2. Is the manuscript technically sound, and do the data support the conclusions?

Reviewer #1: Yes

Reviewer #2: Yes

3. Has the statistical analysis been performed appropriately and rigorously? 

Reviewer #1: Yes

Reviewer #2: Yes

4. Have the authors made all data underlying the findings in their manuscript fully available?

Reviewer #1: Yes

Reviewer #2: Yes

5. Is the manuscript presented in an intelligible fashion and written in standard English?

Reviewer #1: Yes

Reviewer #2: Yes

6. Review Comments to the Author

Reviewer #1: The revised manuscript have shown great improvement and authors have addressed most of the reviewers' comments. The manuscript is ready for publication if authors can address some minor comments as below:

1. Please be careful with typo and some of terms used as they are not accurate. Please make changes to these issues in your revised documents:

Line 141: " stand error" to "standard error"

Line 360: "...evaluated initially" to "...evaluated in this paper..."

Line 361: "...reading length..." to "..read length.."

Line 363: "...sequencing amounts.." to "...sequencing depth..."

Line 363: "...In next..." to "In future research, ,,,"

Line 364: "...whole length..." to "...full length isoform..."

Line 154: change to "...showed it is comprehensive..."

Line 378: sequencing alignment results

2. Line 141: which multiple comparison test is used?

3. "...mRNA fragment randomization test, insert length test, and transcriptome sequencing data saturation test...". Any supporting table or figure for this?

4. The authors mentioned that the description of the objective has been modified to tailor to this study that focus on leaf as the major organ and to generate some initial data. However, I could not find the revised sentence where I expect to have both revision to the objective and justification of using leaf to be in both abstract and last paragraph of introduction. Also, please include the section, paragraph or line information in the authors' response to help the reviewer to identify the revision (at least for the major revision).

5. The conclusion in the abstract and main text body is not specifically linked to the objective of using leaf as sample for the AS study.

Reviewer #2: All my comments have been addressed, only publication number 6 is not correctly named. Must be De Palma M et al.

7. PLOS authors have the option to publish the peer review history of their article (what does this mean?). If published, this will include your full peer review and any attached files.

Reviewer #1: No

Reviewer #2: No

---

## [Author Response · Author response to Decision Letter 1]

21 Oct 2020

Reviewer #1: The revised manuscript have shown great improvement and authors have addressed most of the reviewers' comments. The manuscript is ready for publication if authors can address some minor comments as below:

1. Please be careful with typo and some of terms used as they are not accurate. Please make changes to these issues in your revised documents:

Line 141: " stand error" to "standard error"

Line 360: "...evaluated initially" to "...evaluated in this paper..."

Line 361: "...reading length..." to "..read length.."

Line 363: "...sequencing amounts.." to "...sequencing depth..."

Line 363: "...In next..." to "In future research, ,,,"

Line 364: "...whole length..." to "...full length isoform..."

Line 154: change to "...showed it is comprehensive..."

Line 378: sequencing alignment results

Reply: All were corrected

2. Line 141: which multiple comparison test is used?

Reply: Duncan's multiple range test was used in our experiment. The description was modified in the method section (Statistic analysis).

3. "...mRNA fragment randomization test, insert length test, and transcriptome sequencing data saturation test...". Any supporting table or figure for this?

Reply: The description was from the sequencing company but they did not provide the relative data. Thus, the sentence was removed.

4. The authors mentioned that the description of the objective has been modified to tailor to this study that focus on leaf as the major organ and to generate some initial data. However, I could not find the revised sentence where I expect to have both revision to the objective and justification of using leaf to be in both abstract and last paragraph of introduction. Also, please include the section, paragraph or line information in the authors' response to help the reviewer to identify the revision (at least for the major revision).

Reply: It was modified in line 13-14 in abstract, and last paragraph of introduction.

5. The conclusion in the abstract and main text body is not specifically linked to the objective of using leaf as sample for the AS study.

Reply: It was modified in line 23-25 in abstract, and main text body 

Reviewer #2: All my comments have been addressed, only publication number 6 is not correctly named. Must be De Palma M et al.

Reply: It was corrected.

---

## [Editor Report · Decision Letter 2]

23 Oct 2020

Genome-Wide Profiling of Alternative Splicing Genes in Hybrid Poplar (P.alba×P.glandulosa cv.84K) Leaves

PONE-D-20-17398R2

Dear Dr. Xu,

We’re pleased to inform you that your manuscript has been judged scientifically suitable for publication and will be formally accepted for publication once it meets all outstanding technical requirements.

Kind regards,

Marc Robinson-Rechavi

Academic Editor

PLOS ONE
---

## [Editor Report · Acceptance letter]

3 Nov 2020

PONE-D-20-17398R2 

Genome-Wide Profiling of Alternative Splicing Genes in Hybrid Poplar (*P.alba×P.glandulosa* cv.84K) Leaves 

Dear Dr. Xu:

I'm pleased to inform you that your manuscript has been deemed suitable for publication in PLOS ONE. Congratulations! Your manuscript is now with our production department. 

Kind regards, 

on behalf of

Prof. Marc Robinson-Rechavi 

Academic Editor

PLOS ONE